# Oclacitinib and Myxoma Virus Therapy in Dogs with High-Grade Soft Tissue Sarcoma

**DOI:** 10.3390/biomedicines11092346

**Published:** 2023-08-23

**Authors:** Laura V. Ashton, Kristen M. Weishaar, Bernard Séguin, Amy L. MacNeill

**Affiliations:** 1Microbiology, Immunology and Pathology, College of Veterinary Medicine and Biomedical Sciences, Colorado State University, Fort Collins, CO 80523, USA; laura.ashton@colostate.edu; 2Clinical Sciences, College of Veterinary Medicine and Biomedical Sciences, Colorado State University, Fort Collins, CO 80523, USA; kristen.weishaar@colostate.edu; 3Central Victoria Veterinary Hospital, Victoria, BC V8X 2R3, Canada; bernard.seguin@colostate.edu

**Keywords:** cytokines, oclacitinib, oncolytic virus, dogs, cancer

## Abstract

Human rhabdomyosarcomas are rarely cured by surgical resection alone. This is also true for high-grade soft tissue sarcomas in dogs. Dogs with spontaneous sarcoma are good models for clinical responses to new cancer therapies. Strategic combinations of immunotherapy and oncolytic virotherapy (OV) could improve treatment responses in canine and human cancer patients. To develop an appropriate combination of immunotherapy and OV for dogs with soft tissue sarcoma (STS), canine cancer cells were inoculated with myxoma viruses (MYXVs) and gene transcripts were quantified. Next, the cytokine concentrations in the canine cancer cells were altered to evaluate their effect on MYXV replication. These studies indicated that, as in murine and human cells, type I interferons (IFN) play an important role in limiting MYXV replication in canine cancer cells. To reduce type I IFN production during OV, oclacitinib (a JAK1 inhibitor) was administered twice daily to dogs for 14 days starting ~7 days prior to surgery. STS tumors were excised, and MYXV deleted for serp2 (MYXV∆SERP2) was administered at the surgical site at two time points post-operatively to treat any remaining microscopic tumor cells. Tumor regrowth in dogs treated with OV was decreased relative to historical controls. However, regrowth was not further inhibited in patients given combination therapy.

## 1. Introduction

Dogs that have spontaneous cancers are excellent models of human responses to therapy and directly benefit from new treatment options, as effective treatments are often limited for canine patients [1]. For this study, dogs with spontaneous soft tissue sarcomas were recruited to determine whether oclacitinib (Apoquel^®^, Zoetis, Parsippany, NJ, USA) affected treatment outcomes following surgical excision and post-operative treatment with an oncolytic virus (OV).

Soft tissue sarcomas arise from neoplastic mesenchymal cells within connective tissue components of the body. If complete excision of the STS cannot be achieved, tumor regrowth is expected within approximately one year [2,3]. Frequently, radiation and/or chemotherapy are recommended after surgery to slow the rate of tumor regrowth, but these treatments often do not prevent recurrence [4]. The discovery of alternative therapies, including OV, is important to improve outcomes in patients with STS.

Currently, three OVs are approved for use in people [5,6,7] but no OVs are available for dogs. Oncolytic poxviruses and herpes simplex virus cause cell death in canine sarcoma cells lines and reduce tumor growth rates in mice bearing canine sarcoma xenografts [8,9,10]. Additionally, there is evidence that OVs cause very few adverse clinical events in dogs [11,12,13,14,15,16,17,18,19,20]. These data indicate that OVs could be beneficial to dogs with STS. In most clinical studies, the response to OVs is variable, so OVs are used in combination with other cancer treatments.

It is well established that healthy murine and human cells are stimulated to produce type I IFNs when they are infected with a virus [21,22]. The upregulation of type I IFNs in healthy cells then inhibits viral replication within the infected cells [21]. This is generally believed to happen in many different species and significant amounts of data have shown that this occurs in dogs infected with viruses that are canine pathogens [23]. However, neoplastic murine and human cells often lack an appropriate type I IFN response to virus infection, which allows MYXV replication to occur [24,25]. To our knowledge, the IFN response to virus infection has not been determined in canine cancer cells. Several canine cell cultures were used to evaluate the effects of innate cytokines on MYXV infection, including two soft tissue sarcoma cell cultures.

Type I IFNs have both autocrine and paracrine effects. Secreted type I IFNs bind to the IFNα receptor complex that activates Janus kinase 1 (JAK1) and tyrosine kinase 2 (Tyk2). This activates signal transducer and activator of transcription (STAT) proteins, which translocate to the nucleus. The canonical pathway involves an activated STAT1, STAT2, and IFN-regulatory factor 9 complex. This complex binds to IFN-stimulated response elements in the nucleus and drives the anti-viral cellular response [26]. Alternative type I IFN pathways signal cells through gamma-activated sequences [27] to produce proteins that drive proliferation and could promote tumor invasion, metastasis, and angiogenesis [28]. Theoretically, type I IFN pathway inhibition could reduce both anti-viral responses and cell proliferation in tumors.

Oclacitinib is a JAK1 and JAK2 inhibitor that is used to treat allergic diseases in dogs by reducing serum IL-31 concentration and decreasing pruritis [29]. Interleukin-6 (IL-6) and IL-13 (which signal through JAK1/2 and Tyk2) are inhibited by oclacitinib therapy in dogs [30]. Since type I IFN signaling occurs through the JAK1 and Tyk2 pathways, oclacitinib is also predicted to inhibit IFNα and IFNβ production. Importantly for OV, oclacitinib does not inhibit patient responses to vaccination because its inhibitory activity is relatively poor against JAK3 and Tyk2, which are involved in signaling pathways that drive the T helper 1 (cell-mediated) immune response [29,30,31]. Therefore, this drug could limit type I IFN production and allow viral replication to occur in the tumor for a longer period of time while the virus continues to stimulate an anti-tumoral immune response.

We chose to combine oclacitinib treatment with MYXV∆SERP2 therapy in dogs with STS. Myxoma virus is an oncolytic poxvirus that only causes disease in lagomorphs [32,33,34,35,36,37,38,39] but replicates in murine and human cancer cells that lack appropriate anti-viral IFN responses [24,25]. The data presented in this study indicate that this mechanism also occurs in canine cancer cells. MYXVΔSERP2 was used to treat dogs with STS instead of MYXV because MYXVΔSERP2 is attenuated in rabbits [40] and has improved oncolytic effects in many feline [41] and canine [42] cancer cells. Previous studies indicated that MYXVΔSERP2 treatment is safe in dogs with STS but viral replication lasts < 4 days after injection into tumors [19,43]. We hypothesized that MYXV∆SERP2’s oncolytic effects could be improved by pre-treating canine STS patients with oclacitinib to suppress the anti-viral IFN immune response before MYXVΔSERP2 was administered.

The goal of this study was to determine the effect of cytokines on the replication of MYXV in canine cancer cells. The knowledge gained in cell cultures supported testing the use of oclacitinib (a JAK1 inhibitor) to improve outcomes in dogs with high-grade STS that were treated post-operatively with MYXV∆SERP2. We hypothesized that oclacitinib would inhibit type I IFN responses and prolong MXYV∆SERP2 replication within STS cells that were not surgically removed, thereby limiting the rate of tumor regrowth better than MYXV∆SERP2 treatment alone.

## 2. Materials and Methods

### 2.1. Recombinant Viruses

The Lausanne strain of wild-type MYXV and MYXV expressing tandem dimer tomato red (MYXV-red; vMyx-tdTr) [44] were generously gifted by Dr. Grant McFadden. Myxoma virus deleted for serp2 and expressing lacZ (MYXV∆SERP2, MYXV∆serp2::lacZ) was constructed in the Moyer laboratory [40]. Prior to injection into dogs, MYXVΔSERP2 was sucrose pad-purified, titered, and diluted to 10^6^ foci-forming units (ffu) of MYXV∆SERP2 per mL phosphate-buffered saline (PBS) as previously described [10]. The virus titers were calculated as ffu/mL using standard plaque assays. The plaque assays were performed in 12-well plates (Thermo Fisher Scientific, Nunc, Roskilde, Denmark) containing confluent rabbit kidney epithelial (RK-13, ATTC CCL-37) cells. The cells were inoculated with 10-fold dilutions of samples containing virus, and then, placed in a water-jacketed incubator at 5% CO_2_ and 37 °C for one hour. After incubation, the cells were overlaid with 0.5% agarose (Lonza Group Ltd., Basel, Switzerland) and growth medium consisting of minimal essential medium supplemented with 2 mM glutamine, 50 U/mL penicillin, and 50 µg/mL streptomycin (GE Healthcare, Marlborough, MA, USA); 1 mM sodium pyruvate and 0.1 mM non-essential amino acids (Corning, Corning, NY, USA); and 10% fetal bovine serum (FBS; VWR Life Science Seradigm, Radnor, PA, USA). The cells were incubated for ~5 days, and then, the viral foci that formed in the cell monolayers were visualized and counted. The foci counts were multiplied by the dilution factor and divided by the volume of inoculum to calculate the virus titer (ffu/mL) in the sample.

### 2.2. Canine Cell Cultures

#### 2.2.1. Cell Isolation and Validation

Primary STS cancer cells were derived from spontaneous tumors surgically removed from two dogs that were cancer patients at different Veterinary Teaching Hospitals; STS-1 cells were isolated at the University of Illinois and STS-2 cells were isolated at Colorado State University (CSU). Fibroblasts (FBs) were cultured posthumously from the linea alba of an 11-year-old female dog prior to necropsy at CSU. The cells used in this study were passaged a maximum of 7 times. Data using STS-1 (a.k.a. STSA-1) cells have been published previously and include the effects of MYXV and MYXVΔSERP2 inoculation in cell culture [42], the growth of the cells in a murine xenograft model (confirming their neoplastic potential) [9], the and identification of a frameshift mutation in the neurofibromatosis-1 gene that inactivates neurofibromin, a growth-regulating protein [45,46].

To isolate and culture the cells, a 1 cm^3^ section of tumor or linea was placed in growth medium (described in Section 2.1) in a 35 × 10 mm cell culture dish (Corning, Corning, NY, USA). The tissue was homogenized manually using two 22-gauge needles and placed in a water-jacketed incubator at 5% CO_2_ and 37 °C. The medium was changed daily.

Short tandem repeat (STR) analysis, to ensure each cell culture was unique and of canine origin, and *Mycoplasma* spp. testing were performed at the Cell Line Validation Core at CSU.

Cytochemical staining for vimentin (Clone AP20, BioCare Medical, Pacheco, CA, USA) and cytokeratin (AE1/AE3 + 8/18, BioCare Medical) was performed to validate that the cells in culture expressed proteins that corresponded to the tumor type they were isolated from using previously described methods [42].

#### 2.2.2. Cell Permissivity to MYXV

MYXV-red replication in cells was determined by calculating virus ffu/mL (described in Section 2.1) at several hours post-inoculation (hpi) and constructing virus growth curves [47].

#### 2.2.3. Cytotoxicity of MYXV

The cytotoxic effects of MYXV-red inoculation (0.1 moi) were observed in 48-well plates (Corning, Corning, NY, USA) of confluent cells using a Leica DMI4000B inverted microscope. The expression of the reporter protein tdTr (encoded by MYXV-red) was detected in MYXV-red-inoculated cell cultures using a 560/40 nm bandpass excitation filter associated with the microscope.

Cell viability was measured in canine cells cultured to confluence in 96-well plates (Corning, Corning, NY, USA) and inoculated with MYXV (0.1 moi) using a CellTiter-Glo 2.0 Cell Viability Assay (Promega, Madison, WI, USA) according to the manufacturer’s protocols. At 24, 48, and 72 hpi, cell viability data from mock-infected and MYXV-inoculated cells were compared. The data were analyzed using unpaired, nonparametric t-tests (Mann–Whitney tests) in GraphPad Prism version 9.1.0 software (San Diego, CA, USA).

### 2.3. Evaluation of Cytokine Effects on Viral Infection of Canine Cells

#### 2.3.1. Measurement of mRNAs

##### Canine Immuno-Oncology Panel

An nCounter^®^ Canine IO Panel (NanoString Technologies, Seattle, WA, USA) was used to identify and count transcripts for 800 canine genes (including 20 genes for data normalization). This technology does not require the PCR amplification of transcripts.

The primary canine cells STS-1, STS-2, and FBs were grown to confluency in 6-well plates (Corning, Corning, NY, USA) and inoculated with wild-type MYXV or MYXVΔSERP2 by removing the growth medium (described in Section 2.1) and incubating cells with the virus (0.1 moi) in medium lacking FBS. After incubation in a water-jacketed incubator at 5% CO_2_ and 37 °C for one-hour, the growth medium (described in Section 2.1) was placed on the cells. Cells were harvested for mRNA 4 and 8 hpi using the Monarch Total RNA Miniprep kit (New England Biolabs, Ipswich, MA, USA) according to the manufacturer’s instructions, including the removal of genomic DNA using gDNA Removal Columns provided with the kit.

RNA concentration was determined by measuring absorbance at 260 nm using a 0.5 mm pathlength (ssRNA ng/µL = optical density units (10 mm/0.5 mm) × 40 µg/mL). RNA purity was estimated by calculating the A260/280 and A260/230 ratios. The samples were sent to the University of Arizona Genetics Core (Tucson, AZ, USA) for further evaluation of their concentration and purity and for NanoString analysis. They performed an RNA High-Sensitivity Qubit quantitative read and an RNA High-Sensitivity Fragment Analyzer quality assay, and provided an RNA quality control (QC) summary. They reviewed the NanoString data and reported that “…there are no QC flags and the counts show robustness and some nice diversity. The samples were run targeting between 125 ng and 275 ng in the assay. They were all run with 19 h 65c hybridizations.”

The nanoString data were analyzed using nSolver 4.0 (NanoString Technologies, Seattle, WA, USA) and Rosalind version 3.37.1.0 (Rosalind, San Diego, CA, USA) bioinformatics software.

##### Quantitative PCR

The primary canine cells STS-1 and FBs were grown to confluency in 48-well plates (Corning, Corning, NY, USA) and inoculated with wild-type MYXV or MYXVΔSERP2 by removing the growth medium (described in Section 2.1) and incubating the cells with the virus (0.1 moi) in medium lacking FBS. After incubation in a water-jacketed incubator at 5% CO_2_ and 37 °C for one-hour, growth medium (described in Section 2.1) was placed on the cells. Cells were harvested for mRNA 4 and 24 hpi using the RNeasy^®^ QIAGEN kit according to the manufacturer’s instructions. The optional steps to digest DNA were performed. Moloney murine leukemia virus reverse transcriptase with oligonucleotide (DT_12–18_) primers (Invitrogen, Carlsbad, CA, USA), and 10 mM deoxynucleoside triphosphate (New England Biolabs, Ipswitch, MA, USA) were used to generate cDNA according to the manufacturer’s instructions. Primers (2 µM) against β-actin (housekeeping gene), IFNα, IFNβ, tumor necrosis factor α (TNFα), interleukin-1β (IL-1β), IFNγ, IL-4, IL-10, and transforming growth factor β (TGFβ) were combined with 2× SsoFast™ Evagreen^®^ Supermix (BioRad, Hercules, CA, USA) and 5 µL of sample. Primer sequences are listed in Table 1. Cytokine concentrations were determined using a quantitative PCR (qPCR) thermocycler (Applied Biosystems, QuantStudio, Thermo Fisher Scientific, Waltham, MA, USA) under the following amplification conditions: 94 °C for 1 min, followed by 40 cycles of 95 °C for 15 s, 60 °C for 30 s, a final melting curve at 95 °C for 15 min, and then 60 °C for 15 s. Relative quantification (RQ) values were determined using the ΔΔCT method [48]. Eight data points were recorded per group.

#### 2.3.2. Addition of Cytokines to Canine Cell Cultures

Recombinant canine cytokines were purchased from KingFisher Biotec (St. Paul, MN, USA). These cytokines were used in two separate experiments. First, canine cell cultures were exposed to a range of doses of recombinant cytokines and the cytotoxicity of the cytokines was analyzed. The highest dose of the cytokine that did not cause cytotoxicity in canine cell cultures was determined. Next, cells were treated with the highest dose of cytokine(s) that did not cause cytotoxicity and their effect on tdTr expression in MYXV-red-inoculated canine cell cultures was measured. Details of these two experiments are described below.

Toxicity of the cytokines was evaluated by adding individual and combined recombinant proteins onto 48-well plates (Corning, Corning, NY, USA) containing confluent cells in the absence of virus. The doses of cytokines tested [IFNα (50, 500, and 1000 U/mL), IFNβ (50, 500, and 1000 U/mL), TNFα (10, 20, and 40 ng/mL), and IFNγ (20, 100, and 200 U/mL)] were guided by previous studies [25,51,52,53,54]. A Cell Titer Blue^®^ Cell Viability Assay (Promega, Madison, WI, USA) was used according to the manufacturer’s instructions. Cytotoxicity was calculated as the percentage of viable cells after cytokine treatment relative to untreated cells (fluorescence value of treated cells ÷ fluorescence value of the untreated cells). Data is shown for samples that were run in duplicate. Three similar experiments were performed, although different incubation endpoints were collected. Similar conclusions were made from all experiments.

The highest dose of recombinant cytokine that did not cause toxicity was used to determine if cytokines would alter MYXV-red tdTr expression in canine cells. Confluent cells in 48-well plates (Corning, Corning, NY, USA) were inoculated by removing growth medium (described in Section 2.1.) and incubating cells with MYXV-red at 0.1 moi in medium lacking FBS. After incubation in a water-jacketed incubator at 5% CO_2_ and 37 °C for one-hour, recombinant canine IFNα (500 U/mL), IFNβ (500 U/mL), TNFα (20 ng/mL), and IFNγ (100 U/mL) were diluted in the growth medium (described in Section 2.1.) and placed on the cells alone and in combination. MYXV-red infection in cells was monitored visually by screening for fluorescent red protein expression using a 560/40 nm bandpass excitation filter and a Leica DMI4000B inverted microscope. Reporter protein expression by MYXV-red was measured using a microplate reader by detecting fluorescence (excitation: 554 nm, emission: 581 nm) at 24, 48, and 72 hpi. Four to ten data points at 72 hpi were collected for each canine cell tested. The data were analyzed using multiple unpaired t-tests in GraphPad Prism version 9.1.0 software (San Diego, CA, USA).

#### 2.3.3. Inhibition of Cytokines during MYXV-Red Infection

To block the effects of the cytokine response to viral infection in canine cells, 100 ng of antibody against canine TNFα, IFNα, and IFNγ (KingFisher Biotec, St. Paul, MN, USA) was added alone and in combination to confluent cells infected with MYXV-red for one-hour at 0.1 moi in 48-well plates (Corning, Corning, NY, USA). The antibody against canine IFNβ was not available for purchase. MYXV-red infection in cells was monitored visually by screening for fluorescent red protein expression using a 560/40 nm bandpass excitation filter and a Leica DMI4000B inverted microscope. Virus titers were determined using a plaque assay at 72 hpi. The effect of cytokines on MYXV-red replication was calculated as a percentage of ffu/mL in treated cells relative to untreated cells (ffu/mL of treated cells ÷ ffu/mL of untreated cells). Four to eight data points were collected for each cell line tested. The data were analyzed using unpaired t-tests with Welch’s correction in GraphPad Prism version 9.1.0 software (San Diego, CA, USA).

### 2.4. MYXVΔSERP2 Oncolytic Therapy in Dogs with Sarcomas

#### 2.4.1. Treatment Protocols for Dogs with Sarcomas

Dogs that were otherwise healthy except for the spontaneous growth of a grade II or III soft tissue sarcoma were enrolled in this study as previously described [19]. Briefly, the Clinical Trials Team at CSU Flint Animal Cancer Center oversaw patient enrollment and documented informed client consent. They arranged appointment dates and times, performed physical examinations, and collected the tissue samples used in this study. The post-operative treatment of dogs with MYXVΔSERP2 and combination therapy with oclacitinib (Apoquel^®^, Zoetis, Parsippany, NJ, USA) and post-operative MYXVΔSERP2 (O + MYXV ΔSERP2) were approved by the CSU Veterinary Clinical Studies review board (protocols 2016-016 and 2017-134, respectively). The safety of MYXVΔSERP2 treatment was previously described [19], but clinical outcome data were not available at the time of publication. The outcome data from these dogs are reported in this manuscript (Results Section 3.4.3.).

The administration of the post-operative virus in O+MYXVΔSERP2-treated dogs was performed as previously described [19]. Briefly, sucrose pad-purified MYXVΔSERP2 (5–10 mL) was injected into excised tumor margins at 10^6^ ffu/mL immediately following the surgery (Day 0) and boosted with a second treatment approximately two weeks later. The second treatment was delayed an additional two weeks if the surgical site was not healed or if a seroma was present. Animals enrolled in combination therapy began a 14-day treatment with oclacitinib approximately 7 days prior to surgery. Figure 1 indicates the timeline of sample collections and treatments in dogs with STS. Note that both the post-operative MYXVΔSERP2 and O+MYXVΔSERP2 treatments were pilot clinical trials in client-owned dogs; therefore, the exact timing of treatment varied slightly between patients as appointment times were scheduled around the owners’ availability.

#### 2.4.2. Sample Analysis for Study Dogs

A physical exam, complete blood count (CBC), serum biochemistry profile, and urinalysis were completed for each patient pre-enrollment, and on Days 0, ~14, and ~28, to assess the dogs’ overall health. Three patients also had samples collected ≥ 42 days post-operatively. A board-certified veterinary anatomic pathologist assessed surgically excised tumors for tumor grade, the presence of inflammation, and residual disease. Patients were examined for tumor regrowth monthly for up to ~1 year post-operation.

DNA was extracted from blood, urine, and feces collected from the patients at the time of enrollment (pre-treatment) and on the days specified above. A QIAGEN DNeasy Blood and Tissue kit (Hilden, Germany) was used to isolate DNA according to the manufacturer’s instructions. Detection of the virus was performed using conventional PCR with primers that detect an 818 bp product in the multigene region (M032R, M033R, M034L (DNA polymerase), and M035R) of the MYXV genome (forward 5′CAC CCT CTT TAG TAA AGT ATA CAC C 3′, reverse 5′GAA ATG TTG TCG GAC GGG 3′). A second set of primers covering the M135-M136 genes resulting in an 1182 bp product were also used (forward 5′ CGA CAC CTG TGT ATG TTT G 3′, reverse 5′CCA ATA ACA CAC AGT TCG G 3′). Amplification was performed using a thermocycler under the following conditions: 94 °C for 1 min, followed by 30 cycles of 94 °C for 30 s, 55 °C for 1 min, 72 °C for 2 min, and a final 72 °C 10 min elongation step.

The immune responses of patients to treatment were assessed in several ways. To detect neutralizing antibodies that developed in dogs treated with MXYVΔSERP2, virus neutralization assays were performed using serum collected at the above time points. A plaque reduction neutralization test (PRNT) was used to quantitate anti-viral antibodies as previously described [43,55]. To evaluate subsets of peripheral blood leukocytes, flow cytometric analysis was performed at the Clinical Immunology (Hematopathology) Laboratory at CSU. The numbers of circulating B-lymphocytes (CD21^+^/MHCII^+^), T-lymphocytes (CD3^+^/CD5^+^/CD4^±^/CD8^±^), neutrophils (CD4^+^/CD5^−^), monocytes (CD14^+^/MHCII^+^), and hematopoietic precursors (CD34^+^) were quantified. The data were analyzed using Kaluza 2.2.1 software (Beckman Coulter Life Sciences, Indianapolis, IN, USA). Additionally, the pre-treatment concentrations of serum IL-31 and type I IFNs were compared to the post-treatment concentrations using Nori^®^ Canine ELISA kits (Genorise Scientific, Inc., GlenMills, PA, USA).

The time to tumor regrowth in dogs treated with MYXVΔSERP2 was compared to that of dogs treated with combination therapy (O+MYXVΔSERP2) using an unpaired, nonparametric *t*-test (Mann–Whitney test) in GraphPad Prism software version 9.1.0 (San Diego, CA, USA).

## 3. Results

### 3.1. Characteristics of Canine Cell Lines

Cancer cells from the tumors of two canine patients with spontaneously occurring STS were isolated post-operatively. FBs were collected from one dog post-humorously. All cell cultures were confirmed to be of unique canine lineage using STR analysis. Immunocytochemistry assays were performed to determine whether cells expressed proteins associated with the tumor they were derived from. As expected, all cells expressed vimentin but did not express cytokeratin.

Next, the permissivity of the cells to MYXV-red infection was assessed. The MYXV-red titers increased logarithmically over time in canine cancer cells inoculated with the virus at 0.1 moi, but viral replication in FBs was suppressed (Figure 2). Visible cytotoxic effects were minimal up to 72 hpi with MYXV-red at 0.1 moi in canine STS and FB cells (Figure 3). Cell viability assays collected 24, 48, and 72 hpi with MYXV-red (0.1 moi) indicated that by 24 hpi, there was a significant decrease in the viability of canine STS cells (Figure 4). Although MYXV-red replication (Figure 2) and reporter protein expression were minimal in healthy canine FBs (Figure 3), a significant decrease in FB cell viability was detected at 72 hpi (Figure 4). The data indicate that the canine STS cells are fully permissive to MYXV replication (growth curves are comparable to those in rabbit kidney epithelial cells [56,57]; mean titers = 1.7 × 10^6^ ffu/mL and 1.9 × 10^6^ ffu/mL at 72 hpi for STS-1 and STS-2, respectively) and that healthy canine FBs are poorly permissive to MYXV (mean titer = 6.1 × 10^4^ ffu/mL at 72 hpi).

### 3.2. Cytokine mRNA Expression following MYXV Inoculation

Given that canine STS cells were fully permissive for MYXV infection but MYXV replication was limited in non-cancerous FBs, we predicted that IFN and other cytokine transcription levels would be different in infected STS and FB cells. An nCounter^®^ Canine IO Panel (NanoString Technologies, Seattle, WA, USA) was used to determine how mRNA expression changed in STS-1, STS-2, and FBs following inoculation with MYXV or MYXVΔSERP2 (0.1 moi) as compared to mock-infected cells at 4 and 8 hpi. MYXVΔSERP2 was evaluated because its safety profile in dogs is known [19], and it was used to treat the dogs with spontaneous sarcoma described in this study.

Using nCounter 4.0 Advanced Analysis 2.0 software, heatmaps of key cellular pathways associated with neoplastic transformation and immunologic responses were created. They highlighted several differences between the three mock-infected cell cultures (Figure 5A) and between mock-infected and MYXV- or MYXVΔSERP2-inoculated FB cells (Figure 5B), STS-1 cells (Figure 5C), and STS-2 cells (Figure 5D). Of particular interest for this study, pathway scoring analyses indicated that JAK-STAT and cytokine/chemokine signaling scores were increased in FBs and decreased in STS cells 4 and 8 hpi with viruses (Figure 6). This was expected, given the role of JAK-STAT and cytokine/chemokine signaling in limiting the viral infection of healthy cells, but not in some cancer cells.

Figure 7 shows heatmaps indicating fold changes in selected cytokine transcripts detected in cells using NanoString technology. Cells inoculated with the virus were compared to mock-infected cells collected at the same hpi. IFNα7, TNFSF4, TNFSF13, IL-1β, and IL-10 transcripts were more numerous in all MYXV-inoculated cells at 4 and/or 8 hpi (Figure 7A). This also was observed for IFNβ1 (except in STS-1 cells at 4 hpi and STS-2 cells at 8 hpi) and IL-4 (except in STS-2 cells at 8 hpi) (Figure 7A). IFNγ transcripts remained stable or were down-regulated following MYXV inoculation, with the notable exception of FBs at 8 hpi (Figure 7A). TGFβ1 and TGFβ2 were stable or slightly increased at 4 hpi in all cells, but were down-regulated in STS-1 and STS-2 cells at 8 hpi (Figure 7A). Interestingly, the fold changes in transcript numbers were much more variable following inoculation with MYXVΔSERP2 (Figure 7B). It is possible that the expression of lacZ by this recombinant virus (which is driven by an early/late poxvirus promoter) is the cause for the differences observed in cellular transcript changes when MYXV and MYXVΔSERP2 are compared. Table 2 lists genes with >10-fold differences in expression in MYXV-inoculated as compared to mock-infected FB, STS-1, and STS-2 cells, respectively, at 8 hpi, calculated using Rosalind 3.37.6.0 software. Transcripts that were similarly different in MYXVΔSERP2-inoculated cells as compared to mock-infected cells of the same cell type are indicated by an asterisk. Several similarities in up-regulated and down-regulated genes were detected in MYXV- and MYXVΔSERP2-inoculated cells as compared to mock-infected cells of the same type, although fold changes in MYXVΔSERP2-inoculated cells were often less pronounced than in MYXV-inoculated cells. There was no differential expression detected between mock-infected cells of the same type at 4 and 8 hpi.

Figure 8 shows heatmaps of the RQ values generated using qPCR for IFNα, IFNβ, TNFα, IL-1β, IFNγ, IL-4, IL-10, and TGFβ mRNA expression in STS-1 and FBs following inoculation (0.1 moi) with MYXV (Figure 8A) or MYXVΔSERP2 (Figure 8B) as compared to mock-infected cells. MYXV and MYXVΔSERP2 inoculation caused more similar alterations in transcript levels using this method of detection. At 4 hpi, STS-1 cell mRNA transcripts were relatively stable for all cytokines except IL-4 transcripts, which were increased (Figure 8A,B). An increase in IL-4 transcripts was also observed in FBs inoculated with MYXV (0.1 moi) at 4 hpi (Figure 8A). Virus-inoculated FBs also had a mild increase in IFNα RQ values at 4 hpi (Figure 8A,B).

### 3.3. Effect of Cytokines on MYXV Infection

It was shown that the treatment of primary human FBs with a combination of recombinant human IFNβ and TNF blocked MYXV replication [58]. To determine the effect of adding exogenous cytokines to canine cells following MYXV-red inoculation, canine cells were treated with recombinant canine IFNα, IFNβ, TNFα, and/or IFNγ both alone and in combination. No cytotoxic effects (*p* < 0.05) were observed when cancer cells were treated with the cytokines at the following concentrations: 500 U/mL IFNα, 500 U/mL IFNβ, 20 ng/mL TNFα, and 100 U/mL IFNγ (Figure 9).

When IFNα or IFNβ was added to media following MYXV-red inoculation, MYXV-red reporter protein expression was significantly (*p* < 0.05) decreased in all cells tested (Table 3). Significant reductions in MYXV-red reporter protein expression were also observed in all cells when IFNα and IFNβ were combined with each other, or when either of them was combined with TNFα or with IFNγ. TNFα alone and in combination with other cytokines suppressed reporter protein expression in STS-1 cells but not STS-2 cells or FBs. No significant change in reporter protein expression was observed in cells treated with IFNγ alone.

Similarly, many of the cytokine treatments reduced the virus titers in STS-1 and STS-2 cells (Figure 10). When IFNα was added to media following MYXV-red inoculation, MYXV-red titers were significantly (*p* < 0.05) decreased. IFNβ also decreased MYXV-red titers in STS-2 cells. Significant reductions in MYXV-red titers were observed in STS-1 and STS-2 cells when IFNα and IFNβ were combined with each other, or when either of them was combined with TNFα or with IFNγ (with the exception of STS-2 cells treated with all four cytokines). TNFα alone did not suppress viral replication in STS-1 or STS-2 cells. Somewhat surprisingly, IFNγ alone reduced the viral titers in STS-1 cells.

Conversely, when IFNα, TNFα, and IFNγ were blocked using anti-cytokine antibodies, increases in MYXV-red replication were observed in poorly permissive canine cells (Figure 11). Significant increases (*p* < 0.05) in MYXV-red titers were observed in FBs when the anti-IFNα antibody was used alone or in combination with the anti-TNFα and/or anti-IFNγ antibodies, but not when all three antibodies were combined. This was also seen following treatment with a combination of anti-TNFα and anti-IFNγ antibodies. Interestingly, when treated concurrently with anti-IFNα, anti-TNFα, and anti-IFNγ, MYXV-red replication was decreased in STS-1 cells, which are naturally permissive for viral replication (Figure 2). We speculate that the concurrent inhibition of IFNα, TNFα, and IFNγ promotes the ability of these cells to respond to the other cytokines that are upregulated secondary to MYXV inoculation (including IFNβ and IL-4). This disruption in the balance of intracellular cytokines could lead to a more effective anti-viral immune response. Additional work will need to be conducted to see if this is true.

### 3.4. Post-Operative MYXVΔSERP2 Treatment in Dogs with Soft Tissue Sarcoma

#### 3.4.1. Safety of Combination of Oclacitinib and MYXVΔSERP2 Treatment in Dogs

The safety profile of post-operative MYXVΔSERP2 treatment in five dogs was previously reported [19]. Similar findings were observed in four dogs treated with O+MYXVΔSERP2. Patient information and history are provided in Table 4. No viral shedding was detected in the urine, feces, or blood collected on Days 0, ~14, and ~28, and periodically up to 62 days post-operatively (Appendix A). Also, using Veterinary Comparative Oncology Group (VCOG) criteria [59], no clinically significant changes in physical examination, CBC, serum biochemistry analysis, or urinalysis were detected in dogs treated with O+MYXVΔSERP2 (Appendix A).

#### 3.4.2. Immune Response to Combination of Oclacitinib and MYXVΔSERP2 Treatment in Dogs

Immune responses were evaluated via virus neutralization assays, flow cytometric analysis of the peripheral blood, and the measurement of serum cytokine concentrations. It was previously reported that dogs treated with MYXV∆SERP2 alone did not have pre-existing antibodies that neutralized MYXV (Day 0), nor did they seroconvert on Day 14; however, two of five dogs seroconverted by Day 28 [19]. Likewise, dogs treated with O+MYXV∆SERP2 did not have anti-MYXV antibodies on Day 0 or 14, but two of four dogs seroconverted by Day 28 (Figure 12). Flow cytometric evaluation of peripheral blood leukocytes indicated no significant abnormalities in five of five dogs treated with MYXVΔSERP2 [19] and three of four dogs treated with O+MYXVΔSERP2. One O+MYXVΔSERP2-treated patient (Dog 6) had a low concentration of circulating CD8^+^ T cells (89 cells/µL, reference interval 157–965 cells/µL) on Day 14 (VCOG grade 1; Appendix A). All cell counts were within the reference intervals at the pre-treatment exam and on Days 0, 29, and 43 in this patient. As this finding was transient and was only observed in one patient who had no clinical signs of disease, it was determined to be of no clinical significance.

It is known that oclacitinib inhibits the expression of a variety of cytokines signaled through JAK1 pathways [30]. In dogs treated with oclacitinib, the inhibition of serum IL-31 concentration is indicative of treatment success. Figure 13 compares pre- and post-treatment serum cytokine concentrations in four of five MYXV∆SERP2- and four of four O+MYXV∆SERP2-treated dogs (paired sera from Dog 4 were not available for this assay due to failure to collect a sufficient volume of blood from this patient on Day 0). The interleukin-31, IFNα, and IFNβ concentrations increased in three of four dogs treated with MYXVΔSERP2 alone, whereas the IL-31 and IFNα concentrations remained near the pre-treatment concentrations or decreased in dogs treated with O+ MYXVΔSERP2. The IFNβ concentrations remained near the pre-treatment concentrations or increased slightly in patients treated with combination therapy, but the IFNβ concentrations in all O+MYXV∆SERP2-treated dogs remained lower than the IFNβ concentrations in three of four dogs treated with MYXV∆SERP2 alone. Importantly for this study, the data suggest that oclacitinib successfully suppressed IL-31 and type I IFN responses in O+MYXV∆SERP2-treated dogs.

#### 3.4.3. Clinical Response to Combination of Oclacitinib and MYXVΔSERP2 Treatment in Dogs

We hypothesized that a delayed type I IFN response following MYXV∆SERP2 treatment may allow the virus more time to replicate in tumor cells, thereby improving outcomes in dogs treated with O+MYXV∆SERP2 as compared to dogs treated with MYXV∆SERP2 alone. Tumor regrowth is expected in dogs with STS if the tumor is not completely resected during surgery [2,3]. In dogs treated with MYXV∆SERP2 post-operatively, tumor regrowth was documented in two of four dogs (one dog was lost to follow-up on Day 32). Similarly, in dogs treated with O+MYXV∆SERP2, tumor regrowth was observed in three of four dogs. Table 5 describes the histologic findings in patient tumors and indicates the time to tumor regrowth or the last day of patient follow-up. Clinically, the five patients with tumor regrowth could be classified as having progressive disease [60]. Time to tumor regrowth was not significantly different in dogs treated with MYXV∆SERP2 and dogs treated with O+MYXV∆SERP2.

## 4. Discussion

One goal of this study was to determine whether the manipulation of intracellular cytokine concentrations alters MYXV replication in primary canine cells as it does in murine and human primary cells. When mRNA transcripts were evaluated, the JAK-STAT and cytokine/chemokine pathway signaling scores increased in healthy canine fibroblasts following MYXV and MYXVΔSERP2 inoculation, but decreased in two canine cancer cell cultures (STS-1 and STS-2). The importance of JAK-STAT signaling in the inhibition of viral replication is well established in human cells, including human keratinocyte susceptibility to poxvirus (vaccinia virus) infection [61]. This study suggests that it is also important in canine cells. Moreover, the treatment of cells with type I IFNs reduced MYXV replication in permissive canine cancer cells, and blocking IFNα allowed for more robust MYXV infection in poorly permissive FBs. Specifically, both canine sarcoma cell cultures showed a significant decrease in MYXV replication when cells were treated with IFNα or a combination of IFNα and IFNβ. This is similar to findings in murine cells where blocking type I IFNs allowed for MYXV replication in nonpermissive primary cells [24]. Canine sarcoma cell cultures also showed a significant reduction in MYXV replication in the presence of IFNα or IFNβ when combined with TNFα. The combination of IFNβ and TNFα is known to have a synergistic effect and prevent MYXV infection of human primary cells [25,58].

Previous work demonstrated that MYXV replication is inhibited in canine cells that have low concentrations of phosphorylated Akt (including primary FB) [42]. Blocking IFNα signaling in canine FBs more than doubled MXYV-red titers recovered from the cells at 72 hpi. These new data suggest that defective type I interferon responses in canine cancer cells also are critical in permitting MYXV replication. The similarities observed in human and canine cell culture responses to cytokine alterations support the use of canine cancer patients as models for human cancer patient responses to OVs.

We previously found that oclacitinib treatment prolonged MYXV∆SERP2 replication in rhabdomyosarcoma allografts in mice [62]. Here, we show that the combination of oclacitinib and MYXV∆SERP2 therapy is safe in dogs with STS. However, clinical improvements were not observed in mice or dogs when combination treatment was compared to MYXV∆SERP2 treatment alone. It might be beneficial to combine different type I IFN inhibitors with MYXV therapy. Rapamycin inhibits the mammalian target of rapamycin pathway to decrease the translation of IFNs. The replication of MYXV was increased in cancer cell cultures and in mouse models of melanoma treated with rapamycin [55,63,64,65,66,67,68,69]. Ruxolitinib is a JAK1/2 inhibitor that promotes vesicular stomatitis, herpes simplex, and measles viral replication in cell culture [70,71,72,73,74,75,76]. The drug is not listed for use in dogs; however, a recent study indicated that in vitro canine mast cell proliferation is inhibited by ruxolitinib treatment [77]. The decrease in the proliferation of these cells was associated with the ruxolitinib inhibition of JAK2/STAT5 phosphorylation, indicating that there may be additional benefits to using ruxolitinib to treat canine cancers.

The use of other OVs could improve outcomes in dogs as well. Attenuated adenoviruses have been used to treat canine spontaneous soft tissue sarcomas. A dog with fibrosarcoma was treated with a modified canine adenovirus, and post-treatment surgery successfully removed the tumor [12]. Using a different adenovirus construct, progressive disease was observed in one dog with a schwannoma but a complete response was achieved in one dog with a schwannoma; partial responses were observed in two dogs with hemangiopericytoma; and stable disease was observed in two dogs with fibrosarcomas, two with undifferentiated sarcomas, and one with a hemangiopericytoma [13].

Additional immunotherapeutics that could augment OVs include inhibitors of programed cell death protein 1 (PD-1) or cytotoxic T lymphocyte-associated protein 4 interactions, monoclonal antibodies that block tumor growth, tyrosine kinase inhibitors, cytokines that stimulate inflammation, and pro-apoptotic drugs [78]. One future direction for research in the laboratory is evaluating the effect of concurrent MYXV therapy, PD-1/PD-L1 inhibition, and the cytokine stimulation of macrophages in dogs with spontaneous tumors.

## 5. Conclusions

The inhibition of IFNα increases MYXV replication in healthy canine FBs. Combination treatment with oclacitinib (an inhibitor of type I IFNs) and MYXV∆SERP2 was safe in dogs with STS; however, little treatment benefit was seen; two of four dogs treated with MYXV∆SERP2 and three of four dogs treated with O+MYXV∆SERP2 suffered tumor recurrence. This outcome may be slightly better than that of surgery alone, but larger placebo-controlled clinical trials are needed.

## Figures and Tables

**Figure 1 biomedicines-11-02346-f001:**
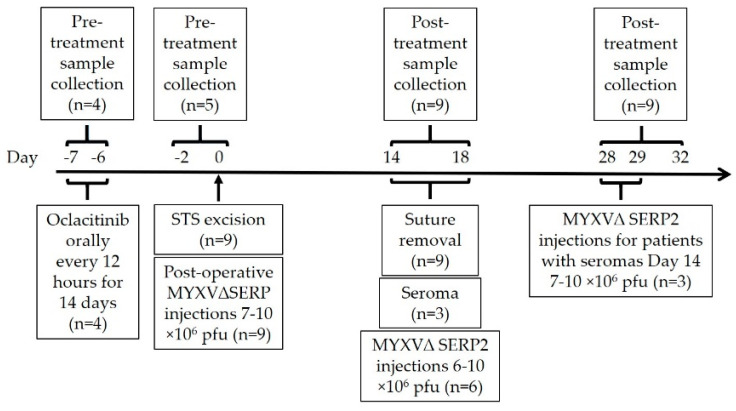
Timeline of sample collections and treatments in dogs with soft tissue sarcoma.

**Figure 2 biomedicines-11-02346-f002:**
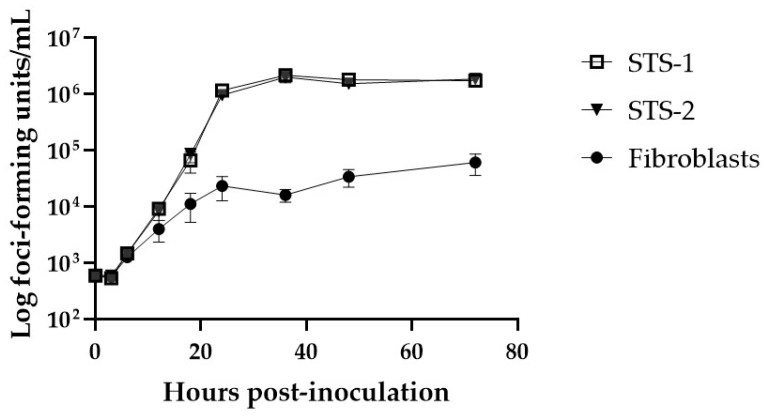
MXYV-red growth rates in canine cells. Viral titers (foci-forming units/mL) in canine soft tissue sarcoma cells (STS-1 and STS-2) and non-cancerous cells (fibroblasts) were measured via plaque assay. Briefly, confluent cells were inoculated with MYXV-red at 0.1 moi; cells were collected at 0, 3, 6, 12, 18, 24, 36, 48, and 72 h post-inoculation; and the viral titer of each sample was determined in RK-13 cells. The graph plots the means and standard error of viral titers in four replicate wells. Significant MYXV-red replication was observed in STS cells.

**Figure 3 biomedicines-11-02346-f003:**
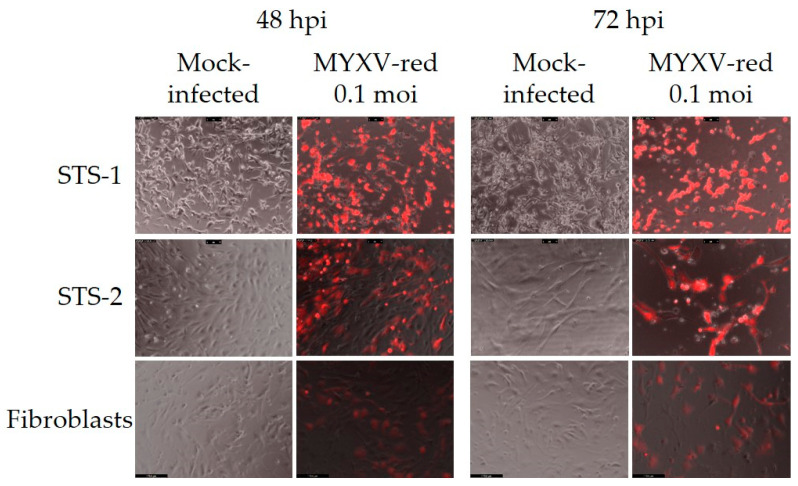
Photomicrographs of representative canine cell cultures (100× magnification). Fluorescent images taken at 48 and 72 h post-inoculation (hpi) with MYXV-red (0.1 moi) are shown for two canine STS cell isolates and canine fibroblasts. (Fibroblasts were isolated from a dog that did not have cancer.) Visible cytotoxic effects were mild even in cancer cells that expressed a significant amount of fluorescent protein.

**Figure 4 biomedicines-11-02346-f004:**
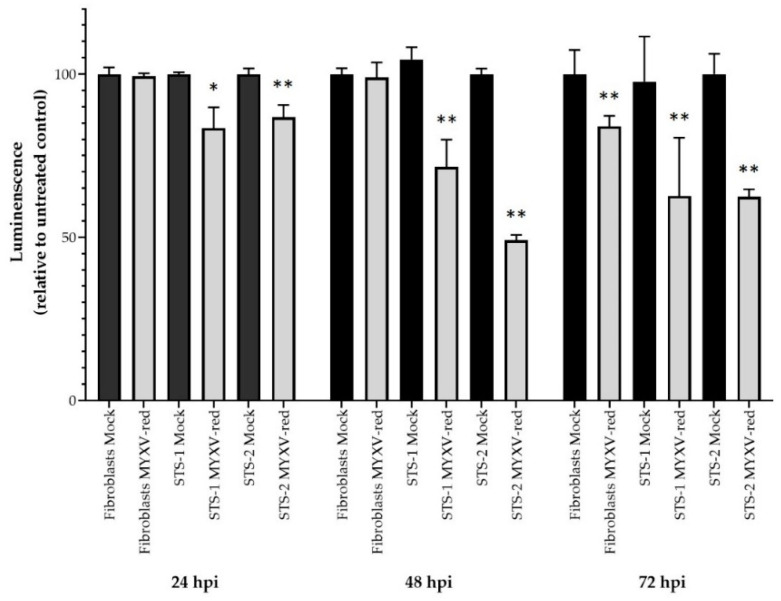
Viability of mock-infected and MYXV-red-inoculated (0.1 moi) canine cells 24, 48, and 72 h post-inoculation (hpi). A CellTIter-Glo Cell Viability Assay (Promega, Madison, WI, USA) was used to detect viable (luminescent) cells. Data were calculated as a percentage of the luminescence signal in untreated cells. Six to thirty replicate wells were averaged. Cell viability was significantly decreased in cancer soft tissue sarcoma cells (STS-1 and STS-2) as early as 24 hpi and in all three cell cultures 72 hpi (* *p* < 0.05, ** *p* < 0.01).

**Figure 5 biomedicines-11-02346-f005:**
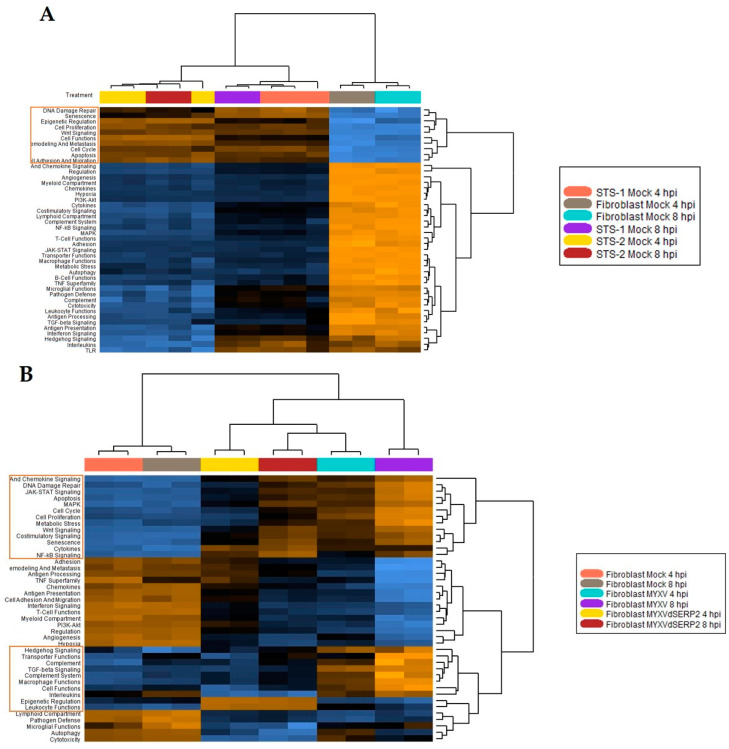
Heatmaps of pathway signatures analyzed using nCounter 4.0 Advanced Analysis 2.0 software. (**A**) Mock-infected FB, STS-1, and STS-2 cells had distinctive pathway signatures as compared to mock-infected FBs at 4 hpi. Pathway signatures that were upregulated in both STS-1 and STS-2 cells are highlighted by the orange box. (**B**) MYXV- and MYXVΔSERP2-inoculated FBs were compared to mock-infected FBs at 4 hpi. Pathway signatures that were upregulated following MYXV or MYXVΔSERP2 inoculation are highlighted. (**C**) MYXV- and MYXVΔSERP2-inoculated STS-1 cells were compared to mock-infected STS-1 at 4 hpi. Pathway signatures that were upregulated following MYXV or MYXVΔSERP2 inoculation are highlighted. (**D**) MYXV- and MYXVΔSERP2-inoculated STS-2 cells were compared to mock-infected STS-2 at 4 hpi. Pathway signatures that were upregulated following MYXV or MYXVΔSERP2 inoculation are highlighted.

**Figure 6 biomedicines-11-02346-f006:**
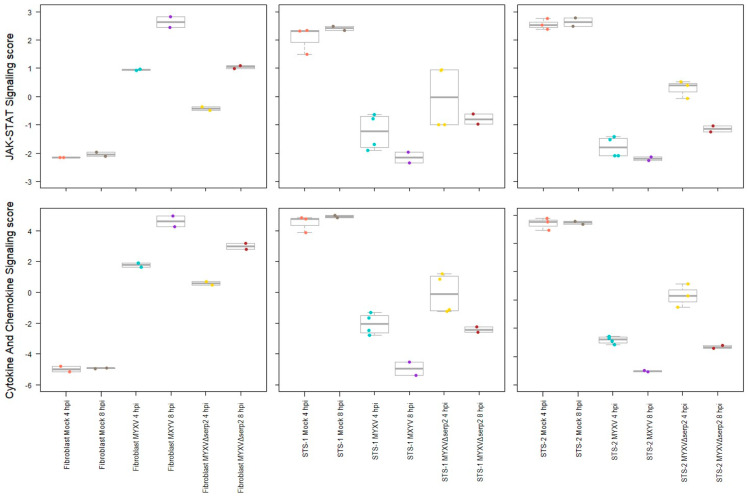
Pathway signaling scores calculated using nCounter 4.0 Advanced Analysis 2.0 software. MYXV- and MYXVΔSERP2-inoculated cells were compared to the same mock-infected cell type at 4 hpi. JAK-STAT and Cytokine/Chemokine Signaling scores were increased in canine FBs following virus infection, but not in sarcoma cells (STS-1 and STS-2).

**Figure 7 biomedicines-11-02346-f007:**
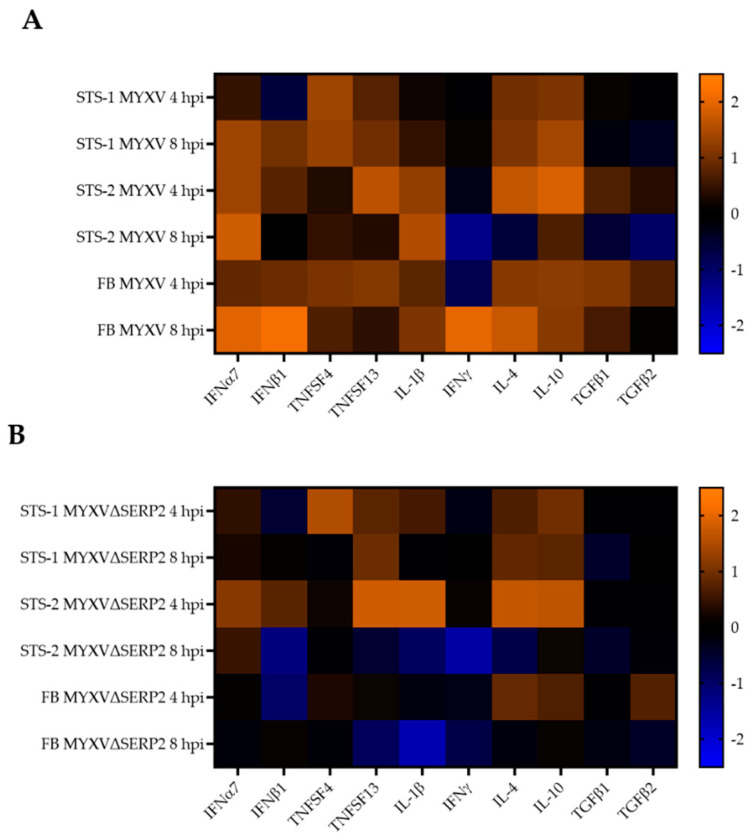
Heatmaps of fold changes in selected cytokine transcripts detected using NanoString technology. The numbers of mRNA transcripts that were detected in cells inoculated with the virus were compared to mock-infected cells collected at the same time points (4 and 8 hpi). (**A**) MYXV-inoculated STS-1 cells and FBs. (**B**) MXYVΔSERP2-inoculated STS-1 cells and FBs.

**Figure 8 biomedicines-11-02346-f008:**
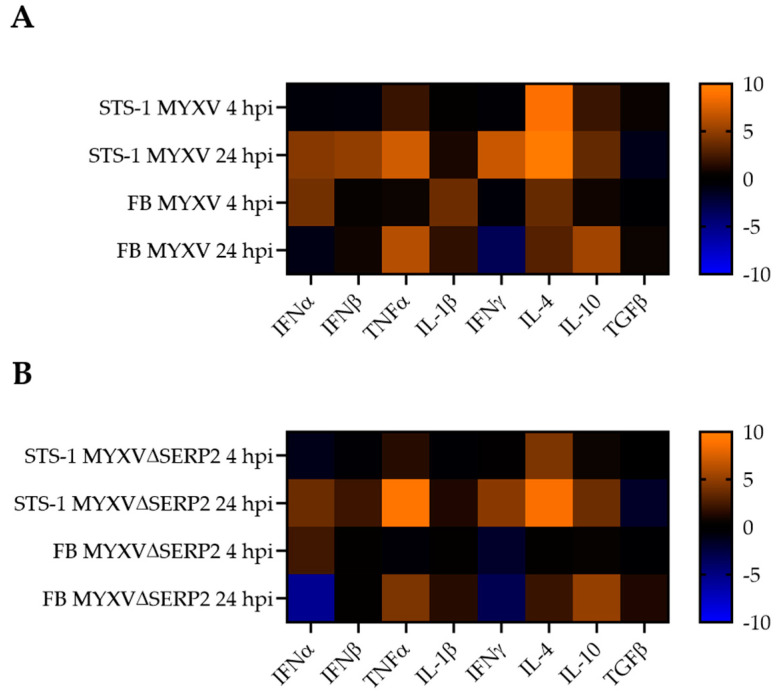
Heatmaps of fold changes in RQ values for selected cytokines detected using qPCR. RQ values from virus-inoculated cells were compared to mock-infected cells collected at the same time points (4 and 24 hpi). (**A**) MYXV-inoculated STS-1 cells and FBs. (**B**) MXYVΔSERP2-inoculated STS-1 cells and FBs.

**Figure 9 biomedicines-11-02346-f009:**
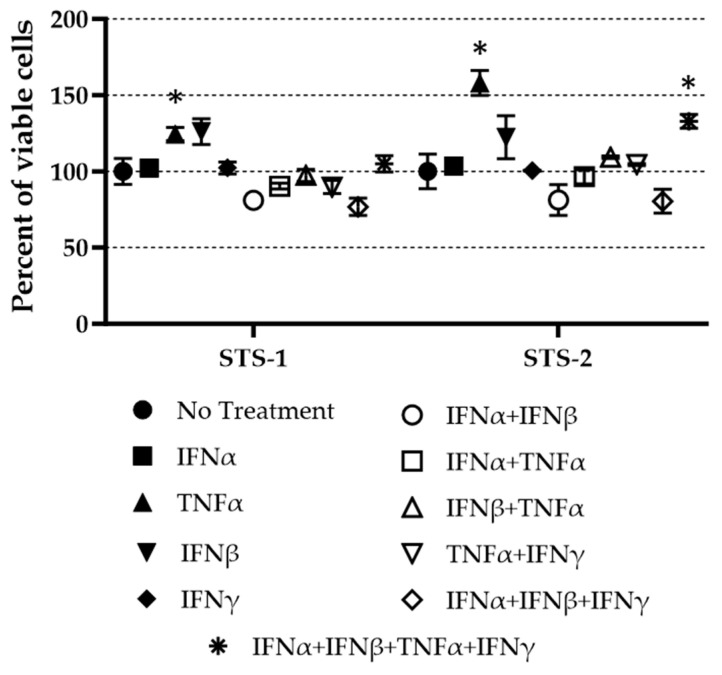
Cytotoxic responses of canine sarcoma cells after 72 h of exposure to cytokines (IFNα (500 U/mL), IFNβ (500 U/mL), TNFα (20 ng/mL), and IFNγ (100 U/mL)). A Cell Titer Blue^®^ Cell Viability Assay (Promega, Madison, WI, USA) was used to detect viable (fluorescing) cells. Data were calculated as a percentage of the fluorescence signal in untreated cells. Experiments were performed a minimum of three times for each cell culture. No statistical decreases in cell viability were observed. * A few statistically significant increases in cell viability were noted (*p* < 0.05).

**Figure 10 biomedicines-11-02346-f010:**
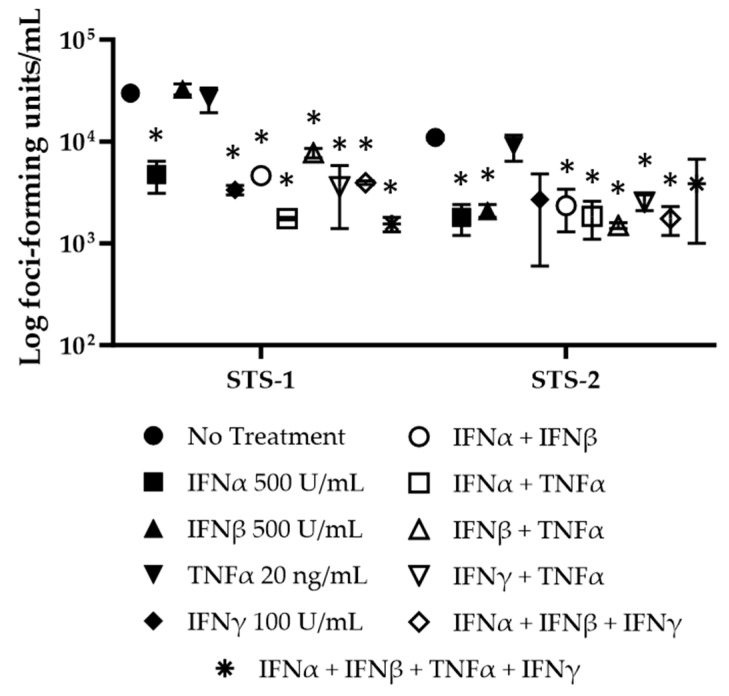
Viral titers from sarcoma cells collected at 72 hpi with MYXV-red (0.1 moi) and treated with recombinant canine cytokines (IFNα (500 U/mL), IFNβ (500 U/mL), TNFα (20 ng/mL), IFNγ (100 U/mL), IFNα (500 U/mL) + IFNβ (500 U/mL), IFNα (500 U/mL) + TNFα (20 ng/mL), IFNβ (500 U/mL) + TNFα (20 ng/mL), TNFα (20 ng/mL) + IFNγ (100 U/mL), IFNα (500 U/mL) + IFNβ (500 U/mL) + IFNγ (100 U/mL), and IFNα (500 U/mL) + IFNβ (500 U/mL) + TNFα (20 ng/mL) + IFNγ (100 U/mL)). * Statistically significant reductions in virus titers were observed (*p* < 0.05).

**Figure 11 biomedicines-11-02346-f011:**
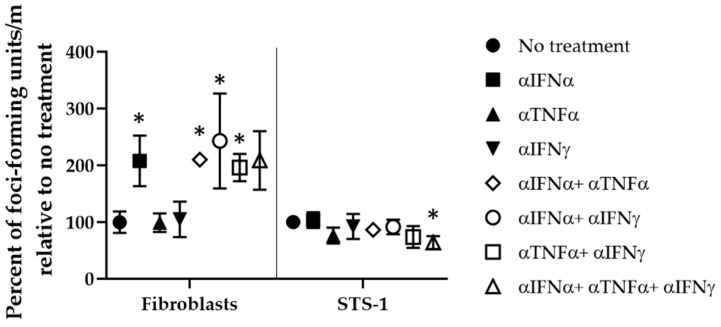
MYXV-red replication (ffu/mL) in canine cells after treatment with antibodies that block cytokine function. * Antibodies (100 ng/mL) against IFNα, TNFα, and/or IFNγ that caused significant changes in MYXV-red replication at 72 hpi (0.1 moi) as compared to untreated cells (*p* < 0.05).

**Figure 12 biomedicines-11-02346-f012:**
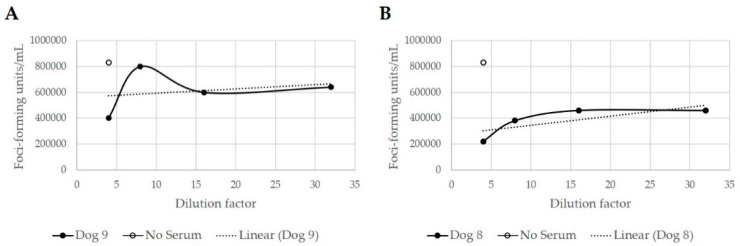
Virus replication following incubation with heat-inactivated canine sera collected on Day 28. (**A**) Data indicating that neutralizing antibodies were not developed in Dog 9 by Day 28 after post-operative treatment with MYXVΔSERP2 was given on Days 0 and 14. (**B**) Data indicating stimulation of neutralizing antibodies in Dog 8 by Day 28.

**Figure 13 biomedicines-11-02346-f013:**
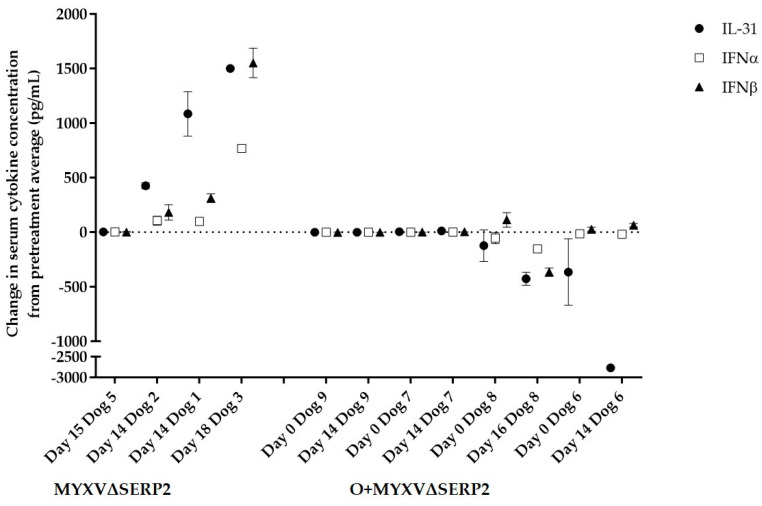
Evaluation of serum cytokine concentrations in dogs treated with MYXV∆SERP2 alone or in combination with oclacitinib (O+MYXV∆SERP2). Post-treatment cytokine concentration results minus pre-treatment results are graphed. Error bars indicate data range. Note that pre-treatment samples were collected prior to surgery (Day 0) for MYXV∆SERP2-treated dogs and prior to oclacitinib treatment (~Day 7) in O+MYXV∆SERP2-treated dogs.

**Table 1 biomedicines-11-02346-t001:** Primer sets used to amplify canine β-actin and cytokine cDNAs in cultured canine cells.

Target cDNA	Forward Primer 5′…3′	Reverse Primer 5′…3′	Reference	GenBank Accession Number	Product Size (Base Pairs)
β-actin	CCG CGA GAA GAT GAC CCA GA	GTG AGG ATC TTC ATG AGG TAG TCG G	[49]	Z70044	81
IFNα	TGG GAC AGA TGA GGA GAC TCT C	GAA GAC CTT CTG GGT CAT CAC G	[50]	AB125936 *	143
IFNβ	CCA GTT CCA GAA GGA GGA CA	TGT CCC AGG TGA AGT TTT CC	[49]	NM_001135787	200
TNFα	GAG CCG ACG TGC CAA TG	CAA CCC ATC TGA CGG CAC TA	[49]	Z70046	79
IL-1β	TCT CCC ACC AGC TCT GTA ACA A	GCA GGG CTT CAG CTT CTC	[49]	Z70047	80
IFNγ	AGC GCA AGG CGA TAA ATG	GCG GCC TCG AAA CAG ATT	[50]	NM_001003174	121
IL-4	CAT CCT CAC AGC GAG AAA CG	CCT TAT CGC TTG TGT TCT TTG GA	[49]	AF054833	83
IL-10	CGC TGT CAC CGA TTT CTT CC	CTG GAG CTT ACT AAA TGC GCT CT	[49]	U33843	78
TGFβ	CAA GGA TCT GGG CTG GAA GTG GA	CCA GGA CCT TGC TGT ACT GCG TGT	[50]	NM_001003309	113

* Primer sequence amplifies products for IFNα subtypes 1 (M28624), 3 (M28626), 4 (AB102731), 5 (AB125934), 6 (NM_001007128), 7 (AB125936), and 8 (AB125937).

**Table 2 biomedicines-11-02346-t002:** Transcripts with >10-fold changes in numbers detected using NanoString technology in MYXV-inoculated canine fibroblasts and sarcoma cells (STS-1 and STS-2) as compared to mock-infected cells collected at 8 h post-inoculation.

Canine Cells	Target	Description	Fold Change
Fibroblasts	PIK3CD	phosphatidylinositol-4,5-bisphosphate 3-kinase, catalytic subunit delta	−67.5
	BRCA1	breast cancer 1, early onset	14.8
	EGR1 *	early growth response 1	31.4
	SERPINB2 *	serpin peptidase inhibitor, clade B (ovalbumin), member 2	69.4
	FOS *	FBJ murine osteosarcoma viral oncogene homolog	82.7
STS-1	SOX10	SRY (sex-determining region Y)-box 10	−68.5
	SELE *	selectin E	10.2
	MCAM	melanoma cell adhesion molecule	10.2
	IL10RA *	interleukin 10 receptor, alpha	12.3
	CTSS	cathepsin S	14.9
	FOS *	FBJ murine osteosarcoma viral oncogene homolog	25.8
	KLRA1	killer cell lectin-like receptor subfamily A, member 1	46.0
	CCL20 *	chemokine (C-C motif) ligand 20	103.5
STS-2	PIK3CD	phosphatidylinositol-4,5-bisphosphate 3-kinase, catalytic subunit delta	−118.5
	CD97	CD97 molecule	−68.0
	CTSS *	cathepsin S	32.5

* Transcripts with >10-fold changes in numbers detected using NanoString technology in MYXVΔSERP2-inoculated canine fibroblasts and sarcoma cells (STS-1 and STS-2) as compared to mock-infected cells collected at 8 h post-inoculation.

**Table 3 biomedicines-11-02346-t003:** Effect of cytokines on MYXV-red reporter protein expression ^†^ in canine cells. Data are expressed as the mean (standard error of the mean) fluorescence units/well relative to no-treatment MYXV-red-inoculated controls.

Treatment	STS-1	STS-2	FBs
No Treatment	100.00 (1.63)	100.00 (1.26)	100.00 (10.28)
IFNα	30.88 * (7.66)	30.10 * (7.97)	40.86 * (5.11)
IFNβ	50.24 * (5.53)	32.44 * (8.67)	17.97 * (5.62)
TNFα	30.02 * (5.06)	134.50 (15.97)	107.05 (14.19)
IFNγ	70.65 (18.42)	97.02 (27.41)	125.57 (6.14)
IFNα + IFNβ	31.12 * (8.57)	22.18 * (5.87)	47.91 * (3.40)
IFNα + TNFα	19.32 * (3.30)	53.54 * (9.08)	55.72 * (5.27)
IFNβ + TNFα	17.69 * (1.58)	36.92 * (7.86)	46.03 * (6.21)
TNFα + IFNγ	23.10 * (5.47)	82.28 (21.85)	113.67 (9.87)
IFNα + IFNβ + IFNγ	24.57 * (5.91)	42.33 * (8.01)	43.71 * (3.86)
IFNα + IFNβ + TNFα + IFNγ	11.59 * (2.13)	27.12 * (6.00)	36.78 * (12.86)

^†^ Fluorescence units were measured at 72 h post-inoculation with MYXV-red at 0.1 multiplicity of infection. * Significant (*p* < 0.05) decrease in MYXV-red reporter gene expression as compared to the no-treatment MYXV-red-inoculated controls. IFNα (500 U/mL), IFNβ (500 U/mL), TNFα (20 ng/mL), IFNγ (100 U/mL). STS (soft tissue sarcoma), FBs (fibroblasts).

**Table 4 biomedicines-11-02346-t004:** Patient information and history.

Patient	Age at Enrollment (Years)	Breed	Sex	Pertinent Medical History Prior to Study Enrollment	Largest Tumor Diameter on Day 0 (cm)	Tumor Location	Histopathology
Dog 1 *	12	Mixed	Spayed female	Tumor excision ~4 months prior Regrowth ~5 months	5 (per lobule)	Multi-lobulated mass, dorsal thorax	Grade 3 undifferentiated sarcoma ^†^
Dog 2 *	11	Greyhound	Castrated male	Tumor excision 2 and 6 years prior Regrowth ~1 month	3.5	Left lateral metatarsus	Grade 2 STS, PNST ^†^
Dog 3 *	15	Labrador retriever	Castrated male	Tumor present 2–3 years	22	Ventral abdomen	Grade 2 STS, myxoid fibrosarcoma ^†^
Dog 4 *	12	Labrador retriever	Spayed female	Tumor excision 2 years prior Regrowth ~1 month	3.5	Right caudolateral thorax	Grade 2 STS, PNST ^†^
Dog 5 *	11	Mixed	Spayed female	Tumor present ~1 month	6.5	Left flank	Grade 2 STS, PNST ^†^
Dog 6	12	Mixed	Castrated male	Tumor excision 6 months prior Regrowth ~3 months	11.5	Right shoulder	Grade 3 STS ^‡^
Dog 7	9	Mixed	Spayed female	Tumor present ~1 month	8	Right antebrachium	Grade 2 STS ^‡^
Dog 8		Rhodesian ridgeback	Spayed female	Tumor excision ~2 years prior; re-excised 1 year prior Regrowth ~1 month	3.9	Left antebrachium	Grade 2 STS ^‡^
Dog 9		Rottweiler	Intact female	Tumor excision 9 months prior Regrowth ~6 months	5	Right shoulder	Grade 3 STS ^‡^

* Table includes some previously published information for dogs treated post-operatively with MYXVΔserp2 (Dogs 1–5 correspond to Dogs 6–10 in the previous publication) [19]. ^†^ Pre-operative biopsy. ^‡^ Post-operative tumor. STS = soft tissue sarcoma. PNST = peripheral nerve sheath tumor.

**Table 5 biomedicines-11-02346-t005:** Tumor histopathology and outcomes in five canine patients with spontaneous soft tissue sarcoma (STS) treated with post-operative MXYVΔSERP2 and four patients treated with a combination of oclacitinib and post-operative MXYVΔSERP2 (O+MXYVΔSERP2).

Patient	Post-Operative Treatment	Excised Tumor Histopathology	Outcome	Post-Operative Day
1	MXYVΔSERP2	Grade III STS, incompletely excised	Recurrence	42
2	MXYVΔSERP2	Grade III STS, incompletely excised	Recurrence	159
3	MXYVΔSERP2	Grade II STS, incompletely excised	Lost to follow-up	32
4	MXYVΔSERP2	Grade II STS, fibrous capsule < 1 mm thick	No regrowth	916
5	MXYVΔSERP2	Grade II STS, incompletely excised	No regrowth, euthanized due to unrelated causes	231
6	O+MXYVΔSERP2	Grade III STS, incompletely excised	Recurrence	188
7	O+MXYVΔSERP2	Grade II STS, incompletely excised	No regrowth, pyogranulomatous inflammation at incision site (Day 31)	378
8	O+MXYVΔSERP2	Grade II STS, incompletely excised	Recurrence	40
9	O+MXYVΔSERP2	Grade III STS, completely excised with 2 mm margins	Recurrence	372

## Data Availability

The data presented in this study are available on request from the corresponding author. The data are not publicly available due to lack of use of public data bases by these authors.

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
