# Peer review of "Oclacitinib and Myxoma Virus Therapy in Dogs with High-Grade Soft Tissue Sarcoma"

_biomedicines, 2023, doi:10.3390/biomedicines11092346_

Round 1

Reviewer 1 Report (Previous Reviewer 4)

The language is fine with extremely minor edits. 

Author Response

Reviewer 2 Report (New Reviewer)

The manuscript entitled “Oclacitinib and myxoma virus therapy in dogs with high grade soft tissue sarcoma” is a well-designed article with very interesting results. The authors performed an in vitro study assessing the effect of a myxoma virus therapy. Please, see some specific comments below:

1.     The cancer cell line characterization was not clear in the manuscript. The authors provided some refences but not specific regarding the cell characterization, genomic stability, etc.   

2.     Line 138. Staining with cytokeratin and vimentin only differentiate mesenchymal from epithelial origin. Do not validate protein produced by the tumor cells.

3.     The clinical section from methods lacks a lot of information. Inclusion criteria, patients’ selection, epidemiological data from the selected animals. Histological subtype, tumor location.

4.     Authors did not provide ethics statement. It is mandatory when use animals.

5.     In results lacks data from the enrolled dogs, including tumor characteristics, location, size…

6.     Authors did not use RECIST for tumor evaluation.

7.     Also, VCOG is not used for evaluation of side effects.

8.     Authors should extensively change the in vivo dada, from methods to discussion.

9.      Provide details regarding toxicity.

Round 2

Reviewer 1 Report (Previous Reviewer 4)

The authors have responded to all my previous concerns and the paper is significantly improved. 

Reviewer 2 Report (New Reviewer)

Have no further comments. 

This manuscript is a resubmission of an earlier submission. The following is a list of the peer review reports and author responses from that submission.

Round 1

Reviewer 1 Report

The Manuscript aims to study the feasibility of combining JAK1-inhibitor together with oncolytic MYXV in dogs with STS. Although the premise of the study is interesting the presented results do not offer any new insight to the field. The manuscript feels incomplete and requires extensive revision and additional experiments before it can be considered for publication.  

Major points:

  1. Materials & Methods section is lacking exact information about the experiments such as: growth media, how cells were plated/seeded for the experiment and exact statistical analysis. Also, the treatment protocol for dogs is not clearly presented. Exact timelines for sample collections and treatments should be clearly indicated (not approximate times).
  2. MYXV cytotoxicity in the presented cell lines must be analysed.
  3. Effect of JAK1-inhibition on MYXV replication/cytotoxicity in the presented cell lines should be analysed.
  4. Figure 2: It is not clear what the analysis indicates. Is the increased fluorescence intensity due to viral spread in the culture? Quantification of infected cells (%) and cell viability should be done.
  5. Table 2: For HSA-1 and HAS-2 treatment with IFNa shows the same numerical value (also with IFNb and TNFa). This indicates data entry error. The table should be re-checked for other errors. As for Figure 2, analysis should be based on percentage of infected and/or killed cells instead of fluorescence units.
  6. Table 2: No explanation/rationale for the used cytokine concentrations is given
  7. Table 2: Data should be presented as a graph instead of table.
  8. Figure 4: The lack of statistical difference is explained by the very low sample amount (according to Methods the samples were run in only as duplicate.). The experiment should be repeated with more samples for statistical analysis.
  9. No explanation is given on why only some of the cell lines were used in Figures 3, 4 and 5.
  10. Figure 5: FB cells don’t produce IFNg in response to MYXV (according to Figure 3). However, anti-IFNg antibody together with anti-TNFa increases MYXV replication. The result is unexpected as anti-TNFa (or anti-IFNg) alone does not enhance MYXV. The Authors should provide at least speculative explanation for these results.
  11. Figure 6: Dog 4 is not in the graph.
  12. The Authors state that “Frequently, radiation and/or chemotherapy are recommended after surgery to slow the rate of tumor regrowth, but these treatments often do not prevent recurrence” How does MYXV∆SERP2 or O+MYXV∆SERP2 compare against radiation or chemotherapy in dogs with STS (recurrence rate, average time to recurrence etc.)? Were radiation/chemotherapy available for the dogs in this trial, or were the dogs selected not eligible for these standard therapies?
  13. “Importantly for OV, oclacitinib does not inhibit patient response to vaccination because its inhibitory activity is relatively poor against JAK3 and Tyk2”

This statement need to be elaborated. Why do the Authors consider JAK3 and Tyk2 especially important in vaccination? The provided references do not support this statement. Also, reference #26 is a “technical monograph” not a peer-reviewed article.

  1. ”Therefore, this drug could limit type I IFN production and allow viral replication to occur in the tumor for a longer period of time while the virus continues to stimulate an anti-tumoral immune response.”

IFN-Is can have pleiotropic effect in the context of oncolytic virotherapy. IFN-I signaling can promote favourable Th1 response leading to enhanced antigen presentation (particularly cross-presentation) and cytotoxic T cell response. On the other hand, it can limit viral replication and promote T cell inhibitory signalling. Therefore, the potential positive and negative effects of JAK-STAT inhibitors (in combination with the oncolytic MYXV) must be discussed in more detail. Especially because no positive synergy was observed in dogs.

  1. “The safety profile of post-operative MYXVΔSERP2 treatment in 5 dogs was previously reported [20].” Are the referenced dogs the same ones that are presented in this manuscript? This should be clearly indicated.
  2. “No viral shedding was detected in in urine, feces, orblood collected on Days 0, ~14, and ~28, and periodically up to 62 days postoperatively. Also, no clinically significant changes in physical examination, CBC, serum chemistry analysis, or urinalysis were detected in dogs treated with O+MYXVΔSERP2.” Data should be included in the manuscript to support these statements.
  3. “Cells from five categories of cancer were isolated post-operatively from the tumors of ten canine cancer patients. FB were collected from one dog post-humorously. All 11 cell cultures were confirmed to be of unique canine lineage using STR analysis. Immunocyto-chemistry was performed to determine if cells expressed proteins associated with the tu-mor they were derived from. As expected, all cells expressed vimentin. MMT-1 was posi-tive for both cytokeratin and vimentin (suggestive of myoepithelial cells).PECAM-1ex-pression was observed  in  both  HSA  cell  cultures,  consistent  with  tumors  derived  from endothelial  Alkaline phosphatase activity was detected in both OSA cell cultures, consistent with osteoblasts.” Data should be included in the manuscript to support these statements.
  4. “Interestingly, when treated concurrently with anti-IFNα, anti-TNFα, and anti-IFNγ, MYXV-red replication was decreased in STS-1cells, which are naturally permissive for viral replication (Figure 2)” This is an unexpected finding and, if true, should be investigated in more detail. Authors should at least provide speculative explanation for these results.
  5. Mentions of analysing seroconversion on day 0 feel really weird. Development of anti-virus antibodies takes time so seroconversion on day 0 can’t be expected. Dogs could of course have pre-existing antibodies.
  6. “Flow cytometric evaluation of peripheral blood leukocytes indicated no significant abnormalities in 5 of 5 dogs treated with MYXVΔSERP2 [20] and 3 of 4 dogs treated with O+MYXVΔ One O+MYXVΔSERP2-treated patient (Patient #6) had a low con-centration of circulating CD8+T cells (89 cells/μL, reference interval 157-965 cells/μL) on Day 14. All cell counts were within reference intervals at the pre-treatment exam and on Days0, 29, and 43in this patient.” Data should be included in the manuscript to support these statements.
  7. “Here we show that combination oclacitinib and MYXV∆SERP2 therapy is safe in dogs with STS.” No data to support this statement is presented.
  8. The Authors should provide evidence of MYXV∆SERP2 infection in the treated dog tumors to support the use this virus.
  9. “It might be beneficial to combine different type I IFN inhibitors with MYXV therapy.” This speculative conclusion is not supported by the results.
  10. Inhibiting IFN-I response can promote viral replication also in healthy tissues. Was there any indication of this? Authors should include analysis of immune responses and viral replication in dogs in response to MYXV therapy.
  11. Not clear what the Authors mean by “manipulation of intracellular cytokine concentrations”. Cytokines were added to the cell culture and act by binding to corresponding cytokine receptor on the cell surface.
  12. “Interestingly, MYXV replication was reduced in all 11 canine cell cultures when only IFNα was present.” This finding is not unexpected, as MYXV is IFN-I sensitive.

Reviewer 2 Report

Interesting manuscript, here are comments for the authors to consider:

Figure 2, Consider including microscopy images at 24hpi relative to 48hpi for a few cell lines. Although florescent intensity increased by 48hpi within the majority of cells, this result doesn’t clearly reflect viral spread. Simply, it could just mean that a lot of RFP was expressed within the initially infected cells, but a significant population remained uninfected.  Was the point of this experiment to validate that these cells are both susceptible and permissive to MYXV infection? If so, would it not also be pertinent to examine how permissive these cells are to MYXVΔSERP2? This could be done by conducting a single-step growth curve examining the replication of MYXV-red virus relative to MYXVΔSERP2 in these cell lines or select candidate models. Consider testing normal canine cells as a control to show that the MYXV-red and MYXVΔSERP2.   Table 2. In some cases, cytokine treatment appears to enhance MYXV-red replication relative to untreated control cells. Interestingly, not only does TNF-or IFN- treatment alone in MMT-2 cells appear to slightly enhance MYXV replication but also enhance cell viability (Figure 4). Could this slight increase in fluorescence units beyond control be due to a greater number of cells in the plate relative to control, which leads to a greater number of infected cells? Given that cell viability measures are not shown for all cell lines (Figure 4) examined in Table 2, it is hard to see if this could be a trend. Consider including the cell viability measure for all cell lines tested.   Figure 5. It would be interesting to see if blocking TNFα, and/or IFNγ influenced MYXV infection in MMT-2 cells. As MYXV infection appears to be enhanced in both MMT-2 and FB cells following TNFα or IFNγ treatment. Considering this, it would be good to include a viability measure for FB cells following cytokine treatment (Figure 4).   Other comments: Why not screen the pharmacoviral combination of oclacitinib with MYXVΔSERP2 (O+MYXV∆SERP2) in the canine cell lines to determine if this approach enhances viral infection in these cells? Particularly, why not try this combination in STS-1 cells and profile the cytokine expression of these cells following treatment. Particularly, test to see if inhibition of IFNβ enhances viral spread (florescent microscopy), replication (single step growth curve), cell killing (cell viability assay) relative to viruses and drug alone.   Despite IFNβ levels of O+MYXV∆SERP2-treated dogs remained lower than 3 of 4 dogs treated MYXVΔSERP2 alone, IFNβ concentrations are reported to remain near pre-treatment concentrations or increased slightly. Moreover, tumor regrowth was found not to be significantly different in O+MYXV∆SERP2 treatment relative to MYXVΔSERP2 alone in dog following STS tumor resection. Considering these results in the context of the preclinical data presented in this paper, I think closer examination of the role of IFNβ in MYXV suppression within STS cells warrants furth investigation.

Reviewer 3 Report

This study aimed at improving the efficacy of oncolytic viruses (namely myxoma virus) to treat dogs with sarcoma. While interesting, the concept is not new, as inhibiting IFN signaling, a major inhibitor of OV efficacy, has been tried multiple times in the past. The methods used are outdated and the results are not clear in terms of efficacy of the stategy. The results in dogs in vivo are interesting, but again, they are at best trends. collectively, I think this work is not suitable for publication and should be rejected. 

Reviewer 4 Report

In the current work, the authors study the ability of innate cytokines (notably IFNa, b, g, and TNF) to inhibit infection with oncolytic MYXV in dogs. The work is relatively timely, in that the ability of IFN responses to inhibit oncolytic infections is being intensively studied, and inclusion of data on how these responses function in dogs would likely advance our understanding of the oncolytic virus/IFN interactions. Unfortunately, despite the potential interest in the direction of the work, numerous technical and conceptual issues are noted with the study that significantly reduce overall enthusiasm. Most notably, the work is based on the hypothesis that inhibition of IFN will increase viral replication within tumors. This hypothesis, however, is directly contradicted by the data shown by the authors in Fig 5 (which shows that inhibition of cytokines improves viral replication in non-malignant fibroblasts, but NOT in malignant cells). This reviewer actually finds this data extremely interesting (in that it is mildly contradictory with some reports for other oncolytic viruses), however, it is difficult to reconcile it with the direction that the manuscript takes. Based on this, the reviewer feels that the author’s might be better served to revisit the overall direction of how they present their data possibly focusing on that fact that IFN responses do NOT influence MYXV replication in dogs. Specific comments are listed below.

The inclusion of references citing Rigvir as being approved for use in people (refs 5 and 6) should probably be revisited. The approval to use this virus has been revoked following wide spread claims of scientific and medical impropriety. While the virus was indeed at one point approved for human use (in Latvia), this reviewer is fairly uncomfortable including these citations as evidence of oncolytic therapies successful application in people.

The use of PFU to describe MYXV infectious units it generally not accurate since MYXV rarely (if ever) forms true virological plaques. This term should likely be changed to foci forming units (FFU) throughout the manuscript.

The description of the experiment shown in Fig 2 should be improved. Based on the text it is unclear that these cells were infected with MXYV and that the increase in fluorescence is supposed to be indicative of viral replication. The text should be altered to clarify these issues.

It is unclear what the data presented in Fig 2 mean biologically. Even non-replicative MYXV infection will produce limited transgene expression from the early/late promoter (as this construct appears to be, based on the citation). Also, it is unclear what percent change in fluorescence means since no back ground level of fluorescence is presented. Because of these issues, it is not really possible to interpret the data in Fig 2 as corresponding to true viral replication. The authors need to provide better data indicating that the cells tested are actually susceptible to MYXV replication.

A table describing the traits of the various cell lines studied would be beneficial.

The experimental design for data shown in Fig 3 should be better described in the text. It is hard to determine what was actually done to generate this data based on the language used in the results section.

Cytokine expression is frequently influenced post-transcriptionally. Simply analyzing the abundance of cytokine mRNA’s can give a very misleading picture of their actual functional expression. For example, TGFB and IL1 both undergo extensive post-translational processing before become functionally secreted. If possible, the work would benefit from a more rigorous analysis of functional cytokine protein levels secreted during infection.

The graphically display in Fig 3 is hard to interpret. This would likely be better shown by separating MXYV and MXYVdSERP2 into separate panels.

The data shown in Table 2 is difficult to meaningfully interpret. MFI of RFP (expressed from MYXV-red) is not indicative of actual viral replication. The authors need to provide stronger evidence that their cytokines actually influence viral replication using more established virological techniques (viral titers, foci forming assays, etc… ).

There appears to be a formatting error on page 10. The first sentence in the paragraph (IFNa (500U/ml….) appears to be a fragment suggestive of missing text (I can’t find the start of the sentence). This issue needs to be corrected.

He authors need to show that their cytokine treatments are functionally effective (induction of some responsive mRNA…).

The data is Fig 5 shows that inhibition of TNF or IFN (or the combination of them all) fails to alter MYXV titer in the two tested malignant cells, but does alter titer in non-malignant fibroblasts. These data are consistent with the previously published fact that many malignant cells have lost the ability to respond properly to IFN and TNF (which is cited by the PI). Unfortunately, while this data is consistent with previous literature, it does NOT support the proposed use of IFN inhibitors in dogs to increase viral replication since this data suggests that the only real impact of that treatment would be to break MYXV’s onco-tropism.

Given the comment above, the results indicating the safety of O+MYXV become relatively important. These data (even though it’s negative) should therefore be included in the paper (at least as supplemental).

The impact of the work would be significantly enhanced by inclusion of some data analyzing the replication of MYXV within the tumor. I’m not sure if this is possible (given that this was a post-surgery study), however, if it is the authors should definitely include this data (IHC, rtPCR, viral titers, etc…)

Author Response

Authors’ Reply to Review Report 4

In the current work, the authors study the ability of innate cytokines (notably IFNa, b, g, and TNF) to inhibit infection with oncolytic MYXV in dogs. The work is relatively timely, in that the ability of IFN responses to inhibit oncolytic infections is being intensively studied, and inclusion of data on how these responses function in dogs would likely advance our understanding of the oncolytic virus/IFN interactions. Unfortunately, despite the potential interest in the direction of the work, numerous technical and conceptual issues are noted with the study that significantly reduce overall enthusiasm. Most notably, the work is based on the hypothesis inhibition of IFN will increase viral replication within tumors. This hypothesis, however, is directly contradicted by the data shown by the authors in Fig 5 (which shows that inhibition of cytokines improves viral replication in non-malignant fibroblasts, but NOT in malignant cells). This reviewer actually finds this data extremely interesting (in that it is mildly contradictory with some reports for other oncolytic viruses), however, it is difficult to reconcile it with the direction that the manuscript takes. Based on this, the reviewer feels that the author’s might be better served to revisit the overall direction of how they present their data possibly focusing on that fact that IFN responses do NOT influence MYXV replication in dogs. Specific comments are listed below.

Thank you very much for your interest in the ideas for this paper.

We think it’s important to think about our findings in the context of a tumor microenvironment, rather than in isolated cell cultures. It is well established in murine and human cells that healthy cells are stimulated to produce Type I IFNs when they are infected with a virus. The IFNs then help to decrease viral replication in the cells.

It has been proposed that the lack of an appropriate Type I IFN response in cancerous cells allows for oncolytic virus replication to occur in cancer cells but not in healthy cells. This has been demonstrated in murine cells. In human cells, it appears a synergistic effect of IFNβ and TNF is needed for to decrease virus replication. To our knowledge, this has not been shown in canine cancer cells.

Our data support the idea that many canine cancer cells fail to express enough Type I IFNs when exposed to virus. Adding IFNα inhibits virus replication in these cells, while blocking IFNα has little effect. Conversely, healthy canine cells block viral replication. It seems that this is due, in part, to IFNα production by these cells. When IFNα is provided to these cells, viral replication is further suppressed. However, when IFNα is blocked, a significant increase in virus titer is seen.

Although most cancer cells do not seem to produce an adequate Type I IFN response to virus infection on their own, it is probable that healthy cells surrounding the cancer cells in a tumor microenvironment do respond to oncolytic virus by secreting Type I IFNs. Type I IFNs could then bind to IFN receptors on the surface of cancer cells and signal them to mount an innate anti-viral response.

We hoped to reduce IFN production by healthy cells in the body (including within the tumor microenvironment) to allow for slightly increased replication of a non-pathogenic oncolytic virus. This could extend the amount of time the host immune system was exposed to viral (and hopefully) tumor antigens. Ideally this would increase the effectiveness of oncolytic viral therapy.

This explanation was shortened and added as the second paragraph of the Introduction. References were provided where indicated. ”Our hypothesis is based on information published in murine and human cells. It is well established that healthy murine and human cells are stimulated to produce type I IFNs when they are infected with a virus [1]. The IFNs then inhibit viral replication in the cells [1]. This is generally believed to happen in many different species and a significant amount of data has shown that this occurs in dogs infected with viruses that are canine pathogens [2]. However, neoplastic murine and human cells often lack an appropriate type I IFN response to virus infection, which allows MYXV replication to occur [3,4]. To our knowledge, this has not been established in canine cancer cells. Several canine cell cultures were used in this study to evaluate the effects of innate cytokines on MYXV infection, including two soft tissue sarcoma cell cultures. Additionally, dogs with spontaneous soft tissue sarcomas were recruited to see if inhibition of type I IFNs affected treatment outcomes following surgical excision and post-operative treatment with an oncolytic virus (OV).”

The inclusion of references citing Rigvir as being approved for use in people (refs 5 and 6) should probably be revisited. The approval to use this virus has been revoked following wide spread claims of scientific and medical impropriety. While the virus was indeed at one point approved for human use (in Latvia), this reviewer is fairly uncomfortable including these citations as evidence of oncolytic therapies successful application in people.

Thank you for this information. I’m embarrassed to say I didn’t realize this. Rigvir references have been removed.

The use of PFU to describe MYXV infectious units it generally not accurate since MYXV rarely (if ever) forms true virological plaques. This term should likely be changed to foci forming units (FFU) throughout the manuscript.

I think it depends on whose poxvirus lab you grew up in, but I have no real preference.  I changed everything to ffu. I also have never seen MYXV create true plaques unless it’s a recombinant virus expressing something lytic.

The description of the experiment shown in Fig 2 should be improved. Based on the text it is unclear that these cells were infected with MXYV and that the increase in fluorescence is supposed to be indicative of viral replication. The text should be altered to clarify these issues. It is unclear what the data presented in Fig 2 mean biologically. Even non-replicative MYXV infection will produce limited transgene expression from the early/late promoter (as this construct appears to be, based on the citation). Also, it is unclear what percent change in fluorescence means since no back ground level of fluorescence is presented. Because of these issues, it is not really possible to interpret the data in Fig 2 as corresponding to true viral replication. The authors need to provide better data indicating that the cells tested are actually susceptible to MYXV replication.

Agreed. We have added a panel (Figure 2A) that shows MYXV-red growth curves (ffu/mL over time) in a subset of the cells. We also added a panel (Figure 2C) to show the direct (albeit weak) correlation between MYXV-red replication (ffu/mL) and MYXV-red reporter protein expression (fluorescence units/well). Also, a new Figure 3 has been added showing photomicrographs of different cell cultures infected with MYXV-red at several time points post-inoculation (moi = 0.1).

A table describing the traits of the various cell lines studied would be beneficial.

A new Table 2 was added.

The experimental design for data shown in Fig 3 should be better described in the text. It is hard to determine what was actually done to generate this data based on the language used in the results section.

Wording was changed slightly. Figure 3 (now Figure 5) “…shows heat maps of differences in RQ values generated using qPCR for IFNα, IFNβ, TNFα, IL-1β, IFNγ, IL-4, IL-10, and TGFβ mRNA expression in STS-1, OSA-1, and FB following inoculation with MYXV or MYXVΔSERP2 (0.1 moi) as compared to mock-infected cells. MYXVΔSERP2 was evaluated because its safety profile in dogs is known [22] and it was used to treat the dogs with spontaneous sarcoma described in this study. The cytokine expression induced by MYXV and MYXVΔSERP2 in each cell culture was similar.”

Cytokine expression is frequently influenced post-transcriptionally. Simply analyzing the abundance of cytokine mRNA’s can give a very misleading picture of their actual functional expression. For example, TGFB and IL1 both undergo extensive post-translational processing before become functionally secreted. If possible, the work would benefit from a more rigorous analysis of functional cytokine protein levels secreted during infection.

Agreed. Unfortunately, the ELISAs we purchased to detect canine cytokines were not sensitive enough to measure cytokine concentrations in any of the cell cultures. Failing to detect the proteins in cell cultures, we switched to quantification of transcripts using qPCR.

The graphically display in Fig 3 is hard to interpret. This would likely be better shown by separating MXYV and MXYVdSERP2 into separate panels.

Done. These are labeled as Figure 5A and Figure 5B in the revised version of the paper.

The data shown in Table 2 is difficult to meaningfully interpret. MFI of RFP (expressed from MYXV-red) is not indicative of actual viral replication. The authors need to provide stronger evidence that their cytokines actually influence viral replication using more established virological techniques (viral titers, foci forming assays, etc… ).

Foci-forming units per cell are now reported for a subset of the data in Figure 7. I agree that these values are not the same, but they are correlated to one another (Figure 2C). Please note that Table 2 has be renamed “Table 3” in the revised manuscript.

Data

There appears to be a formatting error on page 10. The first sentence in the paragraph (IFNa (500U/ml….) appears to be a fragment suggestive of missing text (I can’t find the start of the sentence). This issue needs to be corrected.

This has been corrected. The cytokine concentration information was a Table footnote that somehow became text.

He authors need to show that their cytokine treatments are functionally effective (induction of some responsive mRNA…).

This experiment was not performed. We feel that the changes we show in virus titer and reporter protein expression indicate that the treatments were functionally effective.

The data is Fig 5 shows that inhibition of TNF or IFN (or the combination of them all) fails to alter MYXV titer in the two tested malignant cells, but does alter titer in non-malignant fibroblasts. These data are consistent with the previously published fact that many malignant cells have lost the ability to respond properly to IFN and TNF (which is cited by the PI). Unfortunately, while this data is consistent with previous literature, it does NOT support the proposed use of IFN inhibitors in dogs to increase viral replication since this data suggests that the only real impact of that treatment would be to break MYXV’s onco-tropism.

As mentioned previously, we hypothesize that healthy cells in a tumor microenvironment respond to oncolytic virus by secreting Type I IFNs. In theory, secreted Type I IFNs could act in a paracrine manner to signal cancer cells to mount an innate anti-viral response. By inhibiting IFN production by healthy cells in the body, it is possible that viral tropism will spread to non-cancerous cells and induce unwanted disease. However, the virus we used is highly attenuated even in its natural host, so we felt if a small amount of virus replication in non-cancerous cells occurred, it would be extremely unlikely to cause any pathology. Also, there are additional mechanisms that limit MYXV replication in healthy cells that are not of rabbit origin. One important example of this is that Akt phosphorylation is critical for MYXV replication in murine, human, and canine cells. The hope was that MYXVΔSERP2 would replicate poorly (if at all) in non-cancerous cells while, at the same time, MYXV replication in residual tumor cells would be prolonged until oclacitinib therapy was discontinued. This would increase the amount of time the host immune system was stimulated and increase the effectiveness of MYXV therapy.

Given the comment above, the results indicating the safety of O+MYXV become relatively important. These data (even though it’s negative) should therefore be included in the paper (at least as supplemental).

Supplemental Figure 3 showing PCR results attempting to isolate MYXV DNA in blood, urine, and feces has been added. Clinical information and flow cytometric data are now provided in Supplemental Table 1 and Supplemental Figure 4, respectively.

The impact of the work would be significantly enhanced by inclusion of some data analyzing the replication of MYXV within the tumor. I’m not sure if this is possible (given that this was a post- surgery study), however, if it is the authors should definitely include this data (IHC, rtPCR, viral titers, etc…)

We agree, but we did not think it was appropriate to biopsy the surgical site in these pets.

Round 2

Reviewer 1 Report

The manuscript has somewhat improved. However, it is still not providing any new insight to the field. I cannot recommend this Manuscript to be published.

Major points:

- The data is presented in inconsistent way. This makes reading the manuscript very difficult.

- The effects of JAK1 inhibition is not studied in dog cancer cells in vitro.

- The statements of oclacitinib´s effect on the immune system in combination with MYXV are not substantiated. Discussion of good and bad effects of JAK-STAT signaling (and inhibition of it) in the context of oncolytic viruses should be discussed in detail.

- Authors’ wording “manipulation of intracellular cytokine concentrations” indicates misunderstanding of how the presented in vitro experiments (and cytokines in general) conceptually work.

- The unexpected finding of anti-IFNα, anti-TNFα and anti-IFNγ reducing MYXV-red replication in STS-1 cells should be studied in detail as it contradicts the concept of the work.

Reviewer 2 Report

Reponses are adequate.

Reviewer 4 Report

In the current revision, the authors have added significantly more data to their manuscript which largely addresses most of my previously raised concerns. However, a few points still remain (or were brought up in revision) which should be addressed.

Authors now include new data examining the cytotoxicity of MXYV in various canine cells. This data is presented as % viability, but given the methodology used and the non-lytic tendencies of MYXV it is fairly likely that reductions in the signal obtained are due to cytostasis of infected cultures and not cytolysis. The authors should either provide additional data supporting actual cell death (ie trypan blue staining or flow-based viability dyes) or adjust their labeling and conclusions to take this issue into account.

Fig 7 is mis-labeled as a second Fig 6.

I agree with Reviewer #1 that the paper would benefit from a direct demonstration that oclacitinib enhances viral replication in vitro. While I understand the authors comments that the formulation of oclacitinib prevents this directly, at least a demonstration that another JAK1 inhibitor enhances viral replication (possibly in the presence of exogenous IFN which would presumably be supplied by normal cells in vivo) would make the authors conclusions much stronger.

Display of Fig 7 (labeled as Fig 6) needs to be changed. Simply relying on the order of lines does not seem like a good idea. Maybe this figure can be made to visually be similar to Fig 8?

The authors have now included a comparison of viral titers and TdTR expression. While there is a minor correlation, this data is not really viewed as compelling that TdTR expression accurately measures viral replication. This becomes an issue mostly in Table 3 (which is central to the author’s conclusions). To supplement this data, the authors have now included what is viewed as much better experimental studies on actual viral titers (Fig 6/7). Based on my belief that TdTR is not a meaningful analytical marker, I would recommend conducting a growth curve in FB +/- cytokines and simply removing table 3 (or moving it to supplemental data). Note that if this is done Fig 1B and 1C can likely also be removed or moved to supplemental.

Previously concerns were raised about the basic premise of this work due to the lack of in vitro inhibition of virus in malignant cells. In response to this concern, the authors have altered their introduction which makes the work much clearer. However, they also supply the following statement:

“Our data support the idea that many canine cancer cells fail to express enough Type I IFNs when exposed to virus.”

This conclusion appears to be directly contradicted by the data and conclusions shown in Fig 5 (in which the authors state that “no clear distinctions between the cytokine responses of canine cancer cells and FB inoculated with MYXV or MYXVdSERP2 were detected”). Combined with the data in Table 3 / Fig 7 the authors data appears to suggest that cancerous cells do not secrete different levels of cytokines, and do not respond differently to exogenous cytokines (at least I can’t find any obvious changes in viral inhibition in Table 3). Some discussion about the potential mechanisms involved therefore seems warranted.